# Optimizing Visual Generative Models via Distribution-wise Rewards

**Ruihang Li** [1 2 3]   **Mengde Xu** [3]   **Shuyang Gu** [3]   **Leigang Qu** [4]   **Fuli Feng** [1]   **Han Hu** [3]   **Wenjie Wang** [1]

## Abstract

Conventional reinforcement learning strategies for visual generation typically employ sample-wise reward functions, yet this practice frequently results in reward hacking that degrades image diversity and introduces visual anomalies. To address these limitations, we present a novel framework that finetunes generative models using distribution-wise rewards, ensuring better alignment with real-world data distributions. Unlike rewards that evaluate samples individually, distribution-wise reward accounts for the data distribution of the samples, mitigating the mode collapse problem that occurs when all samples optimize towards the same direction independently. To overcome the prohibitive computational cost of estimating these rewards, we introduce a subset-replace strategy that efficiently provides reward signals by updating only a small subset of a generated reference set. Additionally, we apply RL to optimize post-hoc model merging coefficients, potentially mitigating the train-inference inconsistency caused by introducing stochastic differential equation (SDE) in regular RL practices. Extensive experiments show our approach significantly improves FID-50K across various base models, from 8.30 to 5.77 for SiT and from 3.74 to 3.52 for EDM2. Qualitative evaluation also confirms that our method enhances perceptual quality while preserving sample diversity.

## 1. Introduction

Visual generative models are designed to approximate the complex probability distribution of real-world images and videos. Existing studies have advanced this objective by improving network architectures (Karras et al., 2022; 2024; Chang et al., 2026; Crowson et al., 2024; Wang et al., 2024) and training strategies (Yu et al., 2024b; Huang et al., 2024; Hang et al., 2024). In the post-training stage, reinforcement learning (RL) with sample-wise reward models (Fan et al., 2023; Wu et al., 2023b; Kirstain et al., 2023; Xu et al., 2023; Wang et al., 2025) is employed to align model outputs with human preferences. Nevertheless, reinforcement fine-tuning driven by sample-wise rewards is prone to reward hacking (Weng, 2024; Amodei et al., 2016; Everitt et al., 2017; Gao et al., 2023; Wen et al., 2024; Liu et al., 2025; Li et al., 2025a), often introducing visual artifacts and diminishing the diversity of generated images (Ku et al., 2024; Xue et al., 2025; Miao et al., 2024; Liu et al., 2025), as shown in Figure 1. In contrast, distribution-wise metrics quantify diversity and mode coverage, penalizing generators that miss modes or exhibit low diversity (Borji, 2022; Ku et al., 2024; Cai et al., 2025). Early studies also confirmed their consistency with human judgment and their sensitivity to subtle shifts in the real distribution (Heusel et al., 2017; Borji, 2022), indicating greater robustness compared to sample-wise metrics. Moreover, alignment with a reference distribution that captures holistic, high-level attributes such as visual quality and aesthetics beyond the reach of sample-wise metrics, opens new avenues for improvement.

In this work, we propose a RL approach based on distribution-wise rewards to improve coverage of the real-world data distribution, achieving both high visual fidelity in samples and broad generation diversity. Quantifying the discrepancy between two distributions is a well-studied problem, with established metrics like KL divergence (Joyce, 2011), MMD (Gretton et al., 2006) and Wasserstein distance (Villani, 2009). In the field of image generation, Fréchet Inception Distance (FID) (Heusel et al., 2017; Jayasumana et al., 2024; Chong & Forsyth, 2020) is a widely used metric for assessing the degree of fit between the learned and real image distribution (Karras et al., 2022; 2024; Chang et al., 2026; Crowson et al., 2024; Wang et al., 2024; Yu et al., 2024b; Huang et al., 2024; Hang et al., 2024). Compared to sample-wise metrics like CLIP Score (Hessel et al., 2021) and HPS (Wu et al., 2023b;a), distribution-based metrics provide a better evaluation of how well the generative model covers the real distribution and can identify incorrect fits (Heusel et al., 2017; Gretton et al., 2006; Villani, 2009). As a widely used metric in image generation,

[1]University of Science and Technology of China [2]Shanghai Innovation Institute [3]Hunyuan Frontier Lab, Tencent [4]National University of Singapore. Correspondence to: Wenjie Wang <wenjiewang@ustc.edu.cn>, Leigang Qu <leigangqu@gmail.com>.

*Proceedings of the 43rd International Conference on Machine Learning*, Seoul, South Korea. PMLR 306, 2026. Copyright 2026 by the author(s).

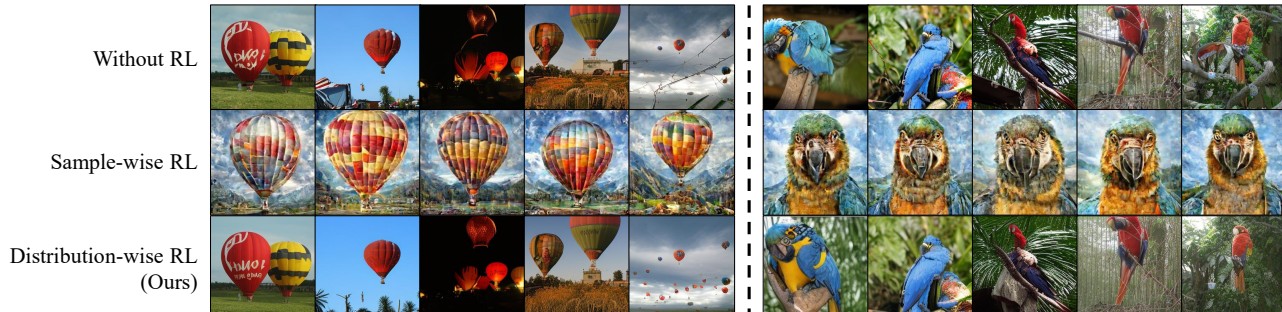

*Figure 1.* Visualization of class-conditional image generation using varied initial noises. The baseline model (without RL, first row, FID 8.30) frequently produces visual artifacts, such as incorrect text rendering, spurious elements, distortion, and vignetting. Applying a sample-wise RL reward [1] leads to severe reward hacking (second row, FID 34.26), causing a collapse in sample diversity and introducing artifacts like bizarre rainbow patterns. In contrast, our **distribution-wise reward (third row, FID 5.77)** significantly mitigates these defects, enhancing overall generation quality and better aligning the learned distribution with the real data.

FID has been validated to correlate well with human perception of visual quality, while also providing a balanced assessment of both fidelity and diversity (Heusel et al., 2017; Salimans et al., 2016; Barratt & Sharma, 2018). Given these advantages, we choose FID as the distribution-wise metric to measure the generative model's fitting capability and use it as the reward signal for reinforcement fine-tuning.

Training with distribution-wise rewards remains underexplored. Existing RL approaches for image generation (Black et al., 2023; Fan et al., 2023; Xue et al., 2025; Liu et al., 2025; Li et al., 2025a) generally treat the denoising process as a Markov Decision Process (MDP) in a stochastic environment (Fan et al., 2023; Liu et al., 2025), employing sample-wise reward models (Fan et al., 2023; Wu et al., 2023b; Kirstain et al., 2023; Xu et al., 2023; Wang et al., 2025) to obtain reward signals for each denoising trajectory, and utilize Group Relative Policy Optimization (GRPO) (Shao et al., 2024; Guo et al., 2025) to optimize the entire state–action sequence. However, directly optimizing with distribution-wise rewards requires computing statistical metrics on a huge set of images (*e.g.*, 50K samples for FID), incurring significant computational cost. Besides, such distribution-wise metrics can't provide reward signals for each individual denoising trajectories that is necessary for RL training. Moreover, we observed that performance improvements from RL fine-tuning in a SDE-based stochastic environment (Fan et al., 2023; Liu et al., 2025; He et al., 2025; Wang & Yu, 2025) for exploration do not fully translate to the faster, ODE-based deterministic sampling used during inference process. This discrepancy highlights a significant train-inference inconsistency and motivates the search for alternative optimization methods that avoids the performance gap between SDE-based training and ODE-based inference.

In this work, we propose distribution-wise reward for RL training. Specifically, we use a novel *subset-replace strategy* to obtain dense distribution-wise reward signals at a low compute cost. First, we generate a reference set of images and compute its FID against the target distribution as a starting point. During rollouts, a small subset of this reference set is replaced by newly generated samples, and the FID of the updated set is used as a dense reward signal. While this signal can be used to directly fine-tune the entire model, and indeed shows promise on models like SiT (Ma et al., 2024), such an approach still requires an SDE-based training formulation (Fan et al., 2023; Liu et al., 2025; Xue et al., 2025; He et al., 2025), inheriting the train-inference inconsistency issue. Inspired by EDM2 (Karras et al., 2024), we explore a more effective optimization strategy: applying our reward signal to search for optimal post-hoc model merging coefficients, instead of fine-tuning all parameters directly. This paradigm decouples the RL optimization from the denoising process, thereby eliminating the potential train-inference gap caused by SDE.

Specifically, the *subset-replace strategy* first computes a base FID on a class-balanced reference set of moderately-sized generated images. During the rollout phase, a small subset ($0.01\times$ of the reference set) of images in the reference set are randomly replaced with newly generated samples of the same corresponding classes. The FID of this partially updated set (replaced FID) is then computed, and its negative value serves as the reward signal for the related subset of images. Experiments on SiT (Ma et al., 2024) demonstrate that our method significantly reduces the FID from 8.30 to 5.77, and the $FD_{DINOv2}$ from 230.39 to 164.88. For post-hoc model merging coefficient optimization, our strategy improves the FID-50K from 3.74 to 3.52 on the EDM2 (Karras et al., 2024) model, highlighting its power as a lightweight,

---

[1]We use ImageReward (Xu et al., 2023) as the sample-wise reward model, formatting prompts as "an image of {class name}" to adapt to the class-conditional generation setting, and

train both the sample-wise and distribution-wise RL models until the reward saturates.

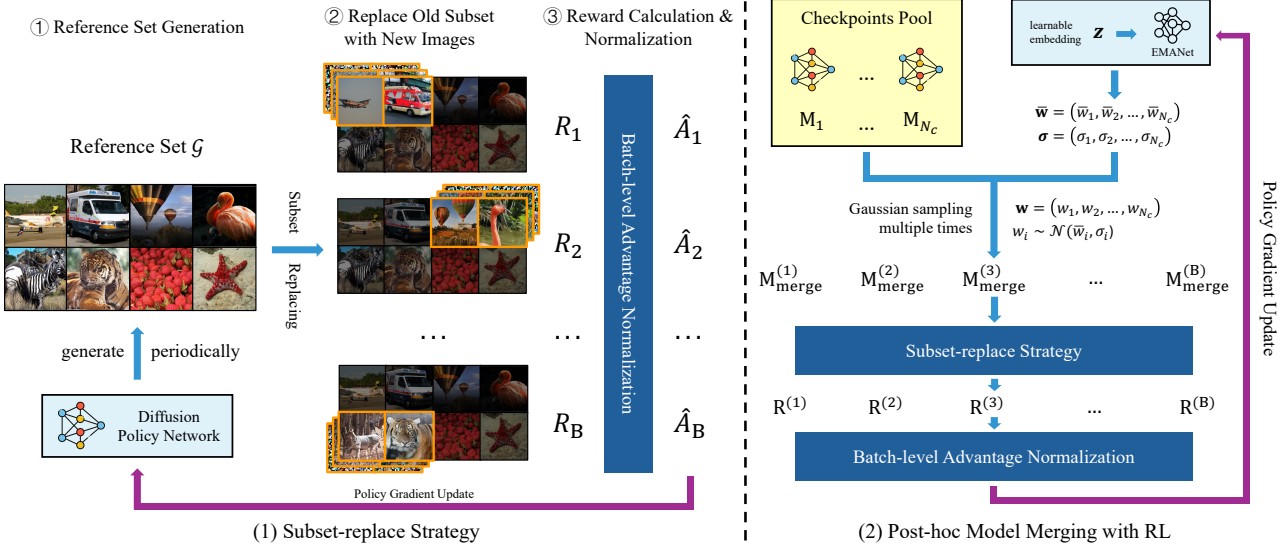

*Figure 2.* Illustration of our proposed RL framework with distribution-wise rewards. **(1) Subset-replace Strategy**: Initially, a reference set is generated using the diffusion policy. During rollout, a random subset is replaced with newly generated samples in the same classes. The distribution-wise metric of the resulting set acts as a reward, which is then normalized into an advantage signal to update the model via policy gradient. The reference set is regenerated periodically. **(2) Post-hoc Model Merging with RL**: The distribution-wise reward signal can guide a lightweight policy to learn the optimal weights for merging a pool of model checkpoints. This efficiently creates an improved model, while allowing the rollout process to utilize ODE-based inference.

plug-and-play module for enhancing pretrained models.

Our contributions are summarized as follows:

1. We analyze the limitations of reinforcement learning with sample-wise reward functions, showing that they are susceptible to reward hacking, which degrades distributional fidelity and introduces artifacts while reducing diversity.

2. We propose a RL framework with distribution-wise rewards by the *subset-replace strategy*. This provides a robust alternative to conventional sample-wise rewards, which are vulnerable to reward hacking. Through extensive experiments, we derive an effective and optimal training recipe that reduces the FID-50K of SiT from 8.30 to 5.77 and the $FD_{DINOv2}$ score from 230.39 to 164.88 without requiring additional training data or architectural modifications.

3. To resolve the train-inference inconsistency of SDE-based RL, we propose a post-hoc optimization of model merging coefficients with distribution-wise reward signals using ODE-based denoising procedure. This training paradigm improves EDM2's FID-50K score from 3.74 to 3.52, validating a more consistent and effective approach to model refinement.

## 2. Related Work

**Reinforcement Learning in Image Generation.** Early works adapted reinforcement learning to diffusion models

by applying policy gradients to the score function (Song et al., 2020), enabling preference-aligned image generation (Black et al., 2023; Fan et al., 2023; Fan & Lee, 2023; Lee et al., 2023). Offline Direct Preference Optimization was later introduced for text-to-image tasks (Wallace et al., 2024), though distributional shift in pairwise data motivated online methods with step-aware preference models (Yuan et al., 2024; Liang et al., 2025). More recently, GRPO-based approaches (Tong et al., 2025; Liu et al., 2025; Xue et al., 2025) have advanced RL-enhanced generation with sample-wise reward models, with (Liu et al., 2025; Xue et al., 2025) extending GRPO to flow matching via ODE–SDE reformulation. (Liu et al., 2025; Xue et al., 2025; Li et al., 2025a) found that reward hacking occurs in the RL process. In this work, we explore the potential to mitigate this issue with distribution-wise rewards. (He et al., 2025; Li et al., 2025a) further employ hybrid SDE–ODE to rollout denoising trajectories to accelerate training. (Wang & Yu, 2025) points out the SDE formulation in common RL practices is injecting greater level of noise than the original ODE, leading to a train-inference inconsistency. In this paper, we applies RL to optimize post-hoc model merging coefficients, eliminating the need for SDE-based rollouts and resolving the train-inference inconsistency of SDE-based RL.

**Distribution-wise Metrics.** Distribution-wise metrics are widely used in training and evaluating neural networks. KL Divergence (Joyce, 2011), which is often included as a regularization term in RL (Fan et al., 2023; Liu et al., 2025; He et al., 2025; Shao et al., 2024; Guo et al., 2025), mea-

sures the difference between distributions but can be unstable when one distribution assigns zero probability to regions where the other has non-zero probability. Maximum Mean Discrepancy (MMD) (Gretton et al., 2006) compares distributions by their means in a Reproducing Kernel Hilbert Space. While non-parametric and robust, MMD can struggle with high-dimensional data and is sensitive to outliers (Lerasle et al., 2019). Frechet Inception Distance (FID) (Heusel et al., 2017), on the other hand, has become the preferred metric to evaluate image generation models (Karras et al., 2022; 2024; Chang et al., 2026; Crowson et al., 2024; Wang et al., 2024; Yu et al., 2024b; Huang et al., 2024; Hang et al., 2024). By comparing feature distributions of real and generated data using a pre-trained Inception network (Szegedy et al., 2016; Heusel et al., 2017), FID reflects how well a generative model fits the real image distribution with lower computational cost and greater statistical robustness. In this work, we introduce a tractable online formulation of the FID, allowing it to be effectively used as a direct distribution-wise reward signal to guide RL in image generation.

**Model Merging.** Model averaging (Izmailov et al., 2018; Polyak & Juditsky, 1992; Tarvainen & Valpola, 2017; Yaz et al., 2018) has become an widely-adopted techniques in the pre-training of state-of-the-art image synthesis models (Balaji et al., 2022; Dhariwal & Nichol, 2021; Ho et al., 2022; Karras et al., 2019; Nichol & Dhariwal, 2021; Peebles & Xie, 2023; Ma et al., 2024; Karras et al., 2022). In the domain of large language models, several studies have similarly explored the use of model averaging during both pre-training (Li et al., 2025b; 2022; Sanyal et al., 2023; Liu et al., 2024; Yang et al., 2023; Dubey et al., 2024; Tian et al., 2025) and post-training (Ilharco et al., 2022; Yu et al., 2024a; Zhou et al., 2024) to improve overall performance and enhance training stability. However, existing approaches such as exponential moving average (EMA) (Morales-Brotons et al., 2024) perform model merging during training, which makes tuning their hyperparameters computationally expensive. (Karras et al., 2024) addresses this limitation by introducing a post-hoc EMA strategy, where the optimal averaging profile is determined through grid search after training. Building on this idea, we propose to optimize the model merging hyperparameters with reinforcement learning, guided by reward signals rather than exhaustive search.

## 3. Method

### 3.1. Preliminaries

**Flow Matching.** Let $\mathbf{x}_0 \sim \mathcal{X}_0$ be drawn from the real data distribution and $\mathbf{x}_1 \sim \mathcal{X}_1$ from a noise distribution. Following the rectified flow framework (Liu et al., 2022),

linear interpolations between the two samples are defined as

$$\mathbf{x}_t = (1-t)\mathbf{x}_0 + t\mathbf{x}_1, \quad t \in [0,1]. \tag{1}$$

A time-dependent velocity field $\mathbf{v}_\theta(\mathbf{x}_t, t)$ is then learned by minimizing the flow-matching objective (Lipman et al., 2022), given by

$$\mathcal{L}_{\text{FM}}(\theta) = \mathbb{E}_{t, \mathbf{x}_0, \mathbf{x}_1}\left[\|\mathbf{v} - \mathbf{v}_\theta(\mathbf{x}_t, t)\|_2^2\right], \quad \mathbf{v} = \mathbf{x}_1 - \mathbf{x}_0. \tag{2}$$

**Denoising as a MDP.** (Black et al., 2023; Liu et al., 2025) cast the iterative denoising procedure in flow matching models as a Markov Decision Process (MDP) $(\mathcal{S}, \mathcal{A}, \rho_0, P, R)$, where $R$ is the reward of this denoising trajectory. Given a class label $c \in \mathcal{C}$, at step $t$, the state is written as $\boldsymbol{s}_t \triangleq (\boldsymbol{c}, t, \boldsymbol{x}_t)$, the action corresponds to the model's prediction $\boldsymbol{a}_t \triangleq \boldsymbol{x}_{t-1}$, and the policy is defined by $\pi(\boldsymbol{a}_t \mid \boldsymbol{s}_t) \triangleq p_\theta(\boldsymbol{x}_{t-1} \mid \boldsymbol{x}_t, \boldsymbol{c})$. The transition is deterministic, *i.e.*, $P(\boldsymbol{s}_{t+1} \mid \boldsymbol{s}_t, \boldsymbol{a}_t) \triangleq (\delta_{\boldsymbol{c}}, \delta_{t-1}, \delta_{\boldsymbol{x}_{t-1}})$, and the initial distribution is specified as $\rho_0(\boldsymbol{s}_0) \triangleq (p(\boldsymbol{c}), \delta_T, \mathcal{N}(\mathbf{0}, \mathbf{I}))$, where $\delta_y$ denotes the Dirac delta distribution centered at $y$.

### 3.2. Subset-Replace Strategy

Existing RL approaches in diffusion models generally formulate the denoising process as a MDP in a stochastic environment (Fan et al., 2023; Liu et al., 2025; Xue et al., 2025; Li et al., 2025a), where a sample-wise reward (Xu et al., 2023; Wang et al., 2025; Wu et al., 2023b; Kirstain et al., 2023) is used as the optimization signal for each denoising trajectory. Directly replacing this with a distribution-wise reward is infeasible: computing such reward typically requires a very large number of trajectories (about 50k images and their denoising trajectories for FID), and assigning the same scalar reward to all trajectories leads to overly sparse feedback, providing little guidance for optimization.

To address these limitations, we propose a **subset-replace strategy** for computing distribution-wise rewards, as demonstrated in Figure 2. Specifically, we first construct a class-balanced moderately-sized reference set $\mathcal{G}$ of $N$ generated images with the initial pretrained model. During rollout, a small subset of $n$ images $g \subseteq \mathcal{G}$ is randomly replaced with newly generated samples $g'$ of the same classes. We then compute the FID of the partially updated set $(\mathcal{G} \setminus g) \cup g'$, denoted as *replaced FID*, whose negative value is used as the reward signal for the associated $n$ denoising trajectories, as shown in Equation 4. To mitigate discrepancies between the reference set and the current model distribution, the reference set is periodically regenerated using the latest model during training. Compared with directly using FID-50K as the reward signal, this strategy substantially reduces computational cost while yielding denser and more informative rewards for model optimization.

*Table 1.* FID (Heusel et al., 2017; Salimans et al., 2016; Barratt & Sharma, 2018) and FD$_{\text{DINOv2}}$ (Stein et al., 2023; Karras et al., 2024) results on ImageNet 256×256. Our results demonstrate that fine-tuning pretrained visual generative models with a distribution-wise reward function is highly effective. This approach significantly enhances the visual quality of generated images within a minimal number of training steps while preserving generative diversity. We validate that the proposed *subset-replace strategy* provides a robust distribution-wise reward signal for both Rejection Sampling (RS) and Policy Gradient Reinforcement Training (RL). Applying our method to a pretrained SiT model reduces the FID-50K score from 8.30 to 6.98 (RS) and 5.77 (RL), validating its efficacy in enhancing perceptual quality. The FD$_{\text{DINOv2}}$ metric exhibits a congruent reduction, confirming the generalizability of this improvement across different feature extractors and demonstrating that our approach is not overfitting to a single metric's feature space.

| Model | Training Steps | FID ↓ | FD$_{\text{DINOv2}}$ ↓ |
|---|---|---|---|
| ADM | 1.98M | 10.94 | - |
| ADM-U | 1.98M | 7.49 | - |
| LDM-8 | 4.8M | 15.51 | - |
| LDM-4 | 178K | 10.56 | - |
| DiT-XL/2 | 400K | 19.50 | - |
| DiT-XL/2 | 7M | 9.60 | - |
| SiT-XL/2 | 400K | 17.20 | - |
| SiT-XL/2 | 7M | 8.30 | 230.39 |
| + Ours (RS) | + 120 | 6.98 | 183.75 |
| + Ours (RL) | + 450 | **5.77** | **164.88** |

We apply the subset-replace strategy to obtain distribution-wise reward signals, and perform direct reinforcement fine-tuning of diffusion models based on them. Following (Fan et al., 2023; Liu et al., 2025), we learn a policy $\pi_\theta$ that maximizes the expected cumulative reward, typically formulated as:

$$\max_\theta \mathbb{E}_{(s_0, a_0, \ldots, s_T, a_T) \sim \pi_\theta}$$

$$\left[ \sum_{t=0}^{T} \Big( R(s_t, a_t) - \beta\, D_{\text{KL}}\big(\pi_\theta(\cdot \mid s_t) \,\|\, \pi_{\text{ref}}(\cdot \mid s_t)\big) \Big) \right]. \tag{3}$$

where the KL-divergence $D_{\text{KL}}$ from a reference policy $\pi_{\text{ref}}$, scaled by $\beta$, serves as a regularization penalty. We adopt a lightweight variant (Shao et al., 2024; Hu, 2025) of traditional policy gradient methods (Schulman et al., 2015; 2017), which estimates the advantage without requiring a value function. Our early experiments presented in Section 4.3 found that batch-level normalization outperforms group-level normalization under our setting, as also observed in (Hu, 2025; Xie et al., 2025).

To formalize the above process, let the reference set $\mathcal{G}$ consist of $N$ generated images. At each iteration, a subset $g$ of $n$ randomly selected images is replaced. Considering rollouts with batch size $B$, the replaced subset is denoted by $\{g_i\}_{i=1}^{B}$, with the corresponding class labels $\{c_i\}_{i=1}^{B}$. We substitute $\{g_i\}_{i=1}^{B}$ with a new subset $\{g_i'\}_{i=1}^{B}$ that preserves the same class distribution, and calculate the reward $R$ as:

$$R(g_i') = -\text{FID}[(\mathcal{G} \setminus g_i) \cup g_i', \overline{\mathcal{G}}], \tag{4}$$

where $\overline{\mathcal{G}}$ denotes the ground-truth image set of the same size as $\mathcal{G}$. Then, the advantage of i-th subset is calculated by:

$$\hat{A}_i = \frac{R(g_i') - \text{mean}(\{R(g_i')\}_{i=1}^{B})}{\text{std}(\{R(g_i')\}_{i=1}^{B})}. \tag{5}$$

Considering the complete denoising trajectory $(x_T^{i,j}, x_{T-1}^{i,j}, \ldots, x_0^{i,j})$ of the $j$-th image in the $i$-th subset, the resulting image subset is given by $g_i' = \{x_0^{i,1}, x_0^{i,2}, \ldots, x_0^{i,n}\}$. Reinforcement fine-tuning then optimizes the policy model $\theta$ by maximizing the following objective as Liu et al. (2025):

$$\mathcal{J}_{\text{Flow-RL}}(\theta) = \mathbb{E}_{c \sim \mathcal{C}, \{x^i\}_{i=1}^{G} \sim \pi_{\theta_{\text{old}}}(\cdot \mid c)} f(r, \hat{A}, \theta, \varepsilon, \beta), \tag{6}$$

where $\pi_{\theta_{\text{old}}}$ is the initial pretrained policy, and

$$f(r, \hat{A}, \theta, \varepsilon, \beta)$$

$$= \frac{1}{B} \sum_{i=1}^{B} \frac{1}{n} \sum_{j=1}^{n} \frac{1}{T} \sum_{t=0}^{T-1} \Bigg($$

$$\min\Big( r_t^{i,j}(\theta)\, \hat{A}_i,\ \text{clip}\big( r_t^{i,j}(\theta), 1 - \varepsilon, 1 + \varepsilon \big) \hat{A}_i \Big)$$

$$- \beta\, D_{\text{KL}}(\pi_\theta \,\|\, \pi_{\text{ref}}) \Bigg),$$

$$r_t^{i,j}(\theta) = \frac{p_\theta(x_{t-1}^{i,j} \mid x_t^{i,j}, c)}{p_{\theta_{\text{old}}}(x_{t-1}^{i,j} \mid x_t^{i,j}, c)}.$$

### 3.3. Post-hoc Model Merging with Distribution-wise Reward

While directly applying our distribution-wise reward signal for fine-tuning with *subset-replace strategy* is a straightforward approach, our experiments in Section 4.3 expose an issue of train-inference inconsistency. Specifically, while existing RL methods on diffusion models (Fan et al., 2023; Xue et al., 2025; Liu et al., 2025) rely on SDEs to provide the stochasticity necessary for the RL process, we observe that the performance gains from this stochastic environment fail to transfer robustly to the ODE-based deterministic samplers (Karras et al., 2022; 2024; Ma et al., 2024) used for standard inference. To bridge this gap, we introduce a post-hoc optimization strategy inspired by EDM2 (Karras et al., 2024). Our method uses RL with distribution-wise rewards to find optimal model merging coefficients, thereby eliminating the dependence on complex SDE solvers (Fan et al., 2023; Liu et al., 2025; Xue et al., 2025) during RL training.

*Figure 3.* Ablation studies on hyperparameters in RL with *subset-replace strategy*. **(a) Reference set size.** The relationship between set size and FID-50K is non-monotonic. While performance generally improves as the size increases from 2,500 to 10,000, the 7,500-sample set exhibits significant degradation, performing worse than even smaller sets. **(b) Number of images to replace.** We evaluate replacing 50, 100, and 200 images in the subset-replace strategy. A smaller replacement size of 50 images yields the best FID-5K performance after 100 training steps. **(c) Impact of rollout sample selection strategies.** Selecting the global top 25% of samples is optimal. Per-process selection methods are inferior, and retaining low-quality samples hinders training.

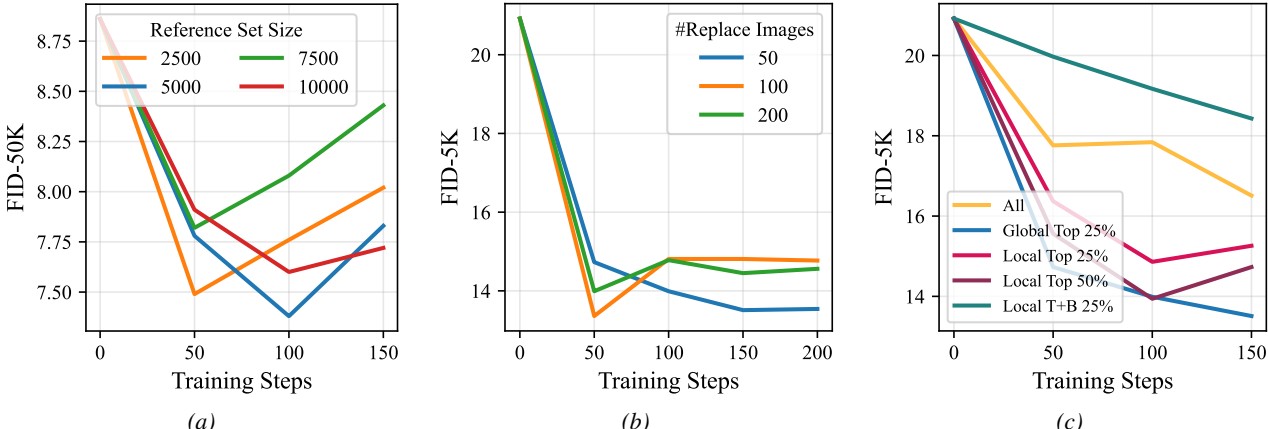

*(a)*        *(b)*        *(c)*

Model merging is a widely used technique in deep learning, and early works in large language models (Li et al., 2025b; Yu et al., 2024a; Zhou et al., 2024) and visual generation models (Balaji et al., 2022; Dhariwal & Nichol, 2021; Ho et al., 2022; Karras et al., 2019; Nichol & Dhariwal, 2021; Peebles & Xie, 2023; Ma et al., 2024; Karras et al., 2022) has demonstrated its effectiveness in stabilizing training and improving model performance. The most common approach is *Exponential Moving Average* (EMA) (Morales-Brotons et al., 2024), which maintains a separate EMA copy of the model and updates it throughout training. However, this requires fixing the merging hyperparameters in advance, often resulting in suboptimal choices. (Karras et al., 2024) shows that by carefully designing the averaging formulation of model replicas during training, it is possible to approximate the EMA version after training. This allows the merging hyperparameters to be adjusted retrospectively based on downstream performance metrics.

To formulate it, let $N_c$ sequential checkpoints along the training trajectory be denoted as $\{M_i\}_{i=1}^{N_c}$, where $M_i$ represents the parameters of the $i$-th model. These checkpoints are then merged into a single final model $M_{\text{merge}}$, where each checkpoint is assigned a weighting coefficient $w_i$. The merged model is computed as:

$$M_{\text{merge}} = \sum_{i=1}^{N_c} w_i M_i \qquad (7)$$

We optimize the model merging coefficients $w_i$ using RL. To introduce the stochasticity and related probabilities required for the RL procedure, we employ a simple MLP policy network $\pi_{\theta_{\text{ema}}}$ (EMANet) to generate the mean $\bar{w}_i$ and standard deviation $\sigma_i$ of each coefficient from a learnable input embedding $z$. The final values $w_i$ are then sampled from a Gaussian distribution

$$w_i \sim \mathcal{N}(\bar{w}_i(z; \pi_{\theta_{\text{ema}}}), \ \sigma_i(z; \pi_{\theta_{\text{ema}}})) \qquad (8)$$

and their corresponding probabilities $p_{w_i}$ are computed as:

$$p_{w_i} = \frac{1}{\sqrt{2\pi\sigma_i^2}} \exp\left(-\frac{(w_i - \bar{w}_i)^2}{2\sigma_i^2}\right). \qquad (9)$$

We regard the coefficients involved in constructing the merged model $M_{\text{merge}}$ as a vector $\mathbf{w} = (w_1, w_2, \dots, w_{N_c})$. The reward corresponding to each $\mathbf{w}$ is computed using the *subset-replace strategy*. During rollouts, we generate a batch of $B$ such coefficient vectors $\{\mathbf{w}^{(j)}\}_{j=1}^{B}$, with the corresponding merged models denoted as $\{M_{\text{merge}}^{(j)}\}_{j=1}^{B}$. For each model $M_{\text{merge}}^{(j)}$, we first construct a reference set $G_j$, from which $N_s$ subsets $\{g_k\}_{k=1}^{N_s}$ are selected. For each subset $g_k$, we replace it with $N_r$ newly generated sets of images $\{g'_{k,p}\}_{p=1}^{N_r}$, obtaining a reward collection

$$\{R_{k,p}^{(j)}\}_{k=1,p=1}^{N_s, N_r}.$$

Finally, the overall reward for coefficient vector $\mathbf{w}^{(j)}$ is defined as the simple average:

$$R^{(j)} = \frac{1}{N_s N_r} \sum_{k=1}^{N_s} \sum_{p=1}^{N_r} R_{k,p}^{(j)}. \qquad (10)$$

We compute the advantages at the batch level (Hu, 2025) across $B$ reward values and use them to update parameters $\theta_{\text{ema}}$ of the policy model. Since the stochasticity in the RL process originates from the coefficient vectors $\mathbf{w}$ generated by $\pi_{\theta_{\text{ema}}}$, it is unnecessary to introduce additional

randomness in the diffusion denoising process. Therefore, we employ efficient ODE sampling (Karras et al., 2022; 2024) throughout the image generation process.

## 4. Experiments

### 4.1. Reinforcement Fine-tuning with Distribution-wise Reward

We use ImageNet (Deng et al., 2009) in 256×256 resolution as our main dataset following (Ma et al., 2025), and perform full parameter reinforcement fine-tuning on SiT (Ma et al., 2024). To lower the training cost, we adopt the denoising reduction technique introduced in (Liu et al., 2025): the number of denoising steps is set to 50 during training and 250 steps during evaluation, following the optimal inference settings in (Ma et al., 2024). We first validated the feasibility of the subset-replace strategy as well as the distribution-wise reward signal under the rejection sampling fine-tuning (RS) setting, and then applied it to the standard RL setting. During RS training, we only use the samples with the highest distribution-wise reward values. Table 1 summarizes FID-50K results of our methods as well as several earlier pretrained models on the ImageNet dataset, following the widely-used evaluation protocol (Karras et al., 2024; Peebles & Xie, 2023; Ma et al., 2024). To demonstrate that our method's efficacy generalizes across different feature representations, we also report $FD_{DINOv2}$ (Stein et al., 2023; Karras et al., 2024) scores. This metric computes the Fréchet Distance of DINOv2 (Oquab et al., 2023) features on 50K ImageNet images, for which we adopt the same evaluation setting from (Karras et al., 2024).

For batch-level advantage normalization, we compute the mean and standard deviation across all processes. In the RL practice, we found that optimization becomes challenging when training on the entire set of rollout samples. To mitigate this, we retain only the top 25% of samples ranked by advantage for parameter update, and further perform detailed ablation experiments in Section 4.3. We adopt an on-policy RL setting in which each rollout sample is used only once for updating the model. Besides, we parallelize reference set generation by distributing tasks across processes and synchronizing the full set to all workers. To balance efficiency and quality, we refresh the reference set with the current model every 10 steps. We performed experiments on 16 NVIDIA Hopper GPUs, and the experiment that yielded the best FID-50K score completed in approximately 20 hours.

Experimental results in Table 1 demonstrate that a simple subset-replace strategy provides an effective distribution-wise reward signal for model optimization. Under the simple RS setting, SiT-XL reduces the FID-50K from 8.30 to **6.98** and $FD_{DINOv2}$ from 230.39 to 183.75, without requiring

*Table 2.* FID results on ImageNet 512×512. The EDM2 baseline results are achieved through post-hoc model merging, with coefficients optimized via extensive grid search, as detailed in (Karras et al., 2024). Results show that using RL to obtain better model merging coefficients is an effective method to boost the performance of pretrained models.

| Model | FID $\downarrow$ |
|---|---|
| ADM (Dhariwal & Nichol, 2021) | 23.24 |
| ADM-U | 9.96 |
| DiT-XL/2 (Peebles & Xie, 2023) | 12.03 |
| EDM2-XS (Karras et al., 2024) | 3.74 |
| + RL-EMA | **3.52** |
| EDM2-S | 2.57 |
| + RL-EMA | **2.52** |

any additional curated training data or architectural modifications. Further incorporating RL, SiT-XL achieves an FID-50K of **5.77** and an $FD_{DINOv2}$ of 164.88 with a small amount of additional training, substantially improving the ability to model image distribution.

### 4.2. Post-hoc Model Merging with Distribution-wise Reward

Following prior settings (Karras et al., 2024), we perform experiments on ImageNet (Deng et al., 2009) (512×512) with models of various sizes to demonstrate the generality of our method. The results are presented in Table 2.

We set $N_c = 8$ to compose the final model $M_{\text{merge}}$. Starting from latest official checkpoints (Karras et al., 2024), we select checkpoints for every $192 \times 2^{20}$ training images, resulting in a checkpoint pool of $N_c = 8$ checkpoints. A simple 3-layer MLP is employed as the policy network to obtain the model merging coefficients $\mathbf{w}$, with the sampling standard deviation fixed to 1.

As shown in Table 2, by optimizing several parameters ($N_c = 8$ in our setting), our method reduces FID from 3.74 to 3.52 on EDM2-XS and from 2.57 to 2.52 on EDM2-S. These results demonstrate that reinforcement learning can effectively optimize model-merging coefficients, yielding further improvements to pretrained models without resorting to complex SDE solvers or training techniques such as denoising reduction (Liu et al., 2025), which has been observed to cause model collapse issues at certain denoising steps.

### 4.3. Ablation Study

We systematically evaluate the influence of key hyperparameters and components in our *subset-replace strategy*, following the experimental protocol in Section 4.1.

*Figure 4.* **Analysis of key design choices for our RL training pipeline.** **(a)** Batch-level advantage normalization for advantages outperforms group-level constantly, yielding faster convergence regardless of whether all or only the top 25% of rollout samples are used for training. **(b)** The performance gap from training-inference inconsistency. A model trained with SDE-based rollouts shows a steadily improving FID score when evaluated with an SDE solver while its performance stagnates when using an ODE solver at the same 250 denoising steps. **(c)** RL training after Rejection Sampling fine-tuning (RS) provided no performance gain, likely due to overfitting from the RS phase. We therefore adopted a pure RL approach.

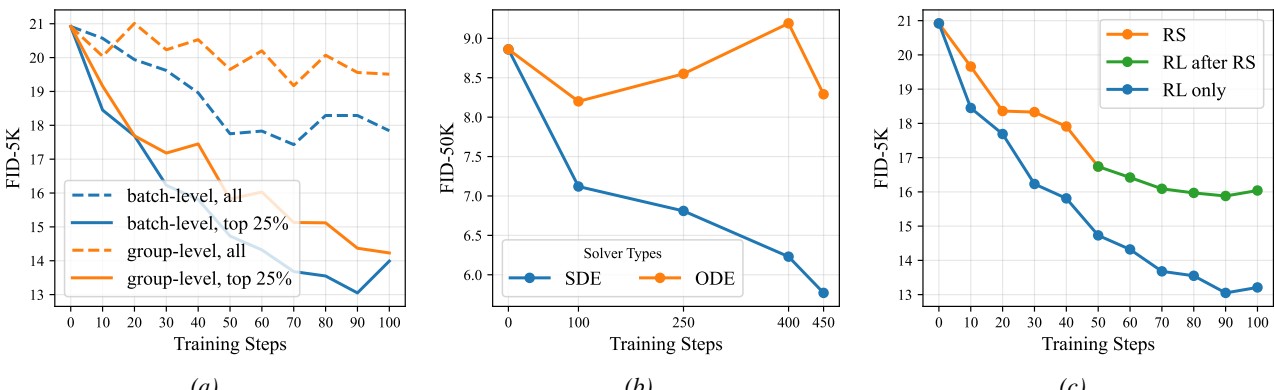

(a)  (b)  (c)

**Advantages normalization.** We compare batch-level and group-level normalization using either all rollout samples or only the top 25% with the highest global advantages (our default setting). As shown in Figure 4a, batch-level normalization consistently yields faster convergence in both settings. Consequently, we adopt batch-level normalization for advantage computation in our final experiments.

**Size of the reference set.** We investigate the trade-off between reward fidelity and computational cost by evaluating reference set sizes of 2,500, 5,000, 7,500, and 10,000. Small sets lack representativeness, while excessively large sets introduce noise, as the replaced batch becomes a statistically insignificant fraction of the total. Figure 3a (reporting FID-50K at 250 steps) reveals a non-monotonic trend: while increasing size from 2,500 to 5,000 improves performance, the 7,500-sample configuration proves unstable. The 5,000-sample set offers the optimal balance between representativeness and stability, which we adopt for our main experiments.

**Number of images to replace during rollout.** The replacement subset size involves a trade-off between signal noise and sparsity. Small subsets are prone to sampling variance, while large ones increase cost and signal sparsity. We tested subset sizes of 50, 100, and 200 with a fixed reference set of 5,000. As shown in Figure 3b, a size of 50 achieves optimal generation quality with the lowest computational overhead, making it our chosen setting.

**Select best samples during RL training.** We analyzed the impact of selecting different sample subsets for training: using all samples, local top 25% or 50% (per process), global top 25%, and top+bottom 25%. Figure 3c demonstrates that the global top 25% setting yields the best perfor-

mance. Retaining lower-quality samples slows convergence, and filtering based on local process rankings proves inferior to the global ranking approach.

**Performance gap between SDE-based training and ODE-based inference.** We employ SDE for rollouts to enable exploration but typically use ODE solvers for efficient inference. Figure 4b reveals that SDE-trained models show negligible gains when evaluated with ODE solvers, highlighting a significant train-inference inconsistency. This aligns with the known dynamic mismatch between SDE and ODE solvers (Deveney et al., 2025). Furthermore, we observe an adaptation bias toward the training denoising schedule under the denoising reduction paradigm (see Appendix A.1 for details), where performance under the training schedule saturates early while the evaluation schedule continues to improve. To resolve this, we propose using RL to optimize post-hoc model merging coefficients, allowing the use of ODE-based rollouts directly during training.

**Pure RL is better than RS-then-RL.** Following common practices in LLMs, we applied the pretrain-SFT-RL paradigm for RL training with distribution-wise reward, where SFT is replaced by reject sampling fine-tuning (RS) in our case. However, the results in Figure 4c indicate that further RL training on the model after RS does not improve performance, likely due to overfitting from the RS phase. Therefore, in the final experiments, we adopted a pure RL setting.

## 5. Conclusion

To address the limitations of sample-wise rewards in RL for visual generation, such as reward hacking and reduced diversity, we propose a novel framework using distribution-wise

rewards enabled by an efficient *subset-replace strategy*. Our method demonstrates significant versatility and effectiveness across multiple scenarios. Through direct fine-tuning, it substantially improves the FID-50K score of SiT from 8.30 to 5.77 and the $FD_{DINOv2}$ score from 230.39 to 164.88. Furthermore, when applied to *post-hoc model merging optimization*, it reduces the FID of EDM2-XS from 3.74 to 3.52 and from 2.57 to 2.52 for EDM2-S, while resolving train-inference inconsistencies in SDE-based RL. These findings validate our approach as an effective method for enhancing the distributional fidelity and perceptual quality of modern generative models.

## Acknowledgments

This work is supported by the National Natural Science Foundation of China (U25B2071).

## Impact Statement

This paper presents work whose goal is to advance the field of Machine Learning, specifically in improving the alignment and fidelity of visual generative models. By introducing a distribution-wise reinforcement learning framework, our approach effectively mitigates the "reward hacking" phenomenon often observed in sample-wise RL, which tends to diminish generation diversity. From a societal perspective, maintaining high diversity and mode coverage is crucial for ensuring that generative models fairly represent the full spectrum of the underlying data distribution, rather than collapsing to a few dominant modes. While improvements in photorealism carry inherent risks regarding potential misuse for misinformation or deepfakes, our work focuses on aligning models more faithfully to the reference data distribution without introducing external biases or artifacts. We believe that robust, distribution-preserving alignment techniques are essential for developing reliable and representative generative systems.

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

# A. Hyperparameter Details

Our model is fine-tuned using the Adam optimizer ($\beta_1 = 0.9, \beta_2 = 0.999$, no weight decay) with a constant learning rate of $1 \times 10^{-5}$. During policy gradient updates, rollouts are performed with global batch size of 128, and the KL-divergence regularization scaler $\beta$ is set to 0. The policy network is updated once per rollout step with a global batch size of 128.

## A.1. More Ablation Studies

**Reference Set Refresh Interval.** In training with the subset-replace strategy, the reference set is periodically regenerated by the current model after a fixed number of steps. Large intervals cause the reference set to lag behind, reducing reward representativeness, while small intervals incur unnecessary overhead. We conduct ablation experiments with intervals of 5, 10, and 20, using the FID-5K of the reference set as the evaluation metric. As shown in Figure 5, an interval of 10 achieves the best final generation performance while maintaining a balanced computational cost.

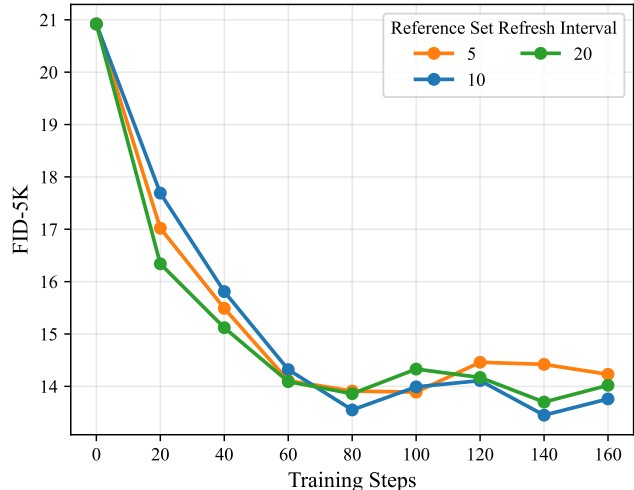

*Figure 5.* Ablation results on reference set refresh interval. We compare intervals of 5, 10, and 20 training steps, finding that 10 steps achieves the best FID-5K score by providing a good balance between reward representativeness and computational overhead.

**Adaptation bias toward the training denoising schedule.** We observed that after the model reaches its optimal performance, its performance gradually deteriorates as RL training continues. Our experiments suggest that this phenomenon is not due to general overfitting, but rather an adaptation bias toward the specific denoising schedule used during training under the denoising reduction paradigm (Liu et al., 2025). In the setup described in Section 4.1, the model adopts 50 denoising steps during training to generate a reference set of 5k images for FID-5K@50, while evaluation uses 250 denoising steps on 50k images for FID-50K@250. We also measured FID-5K@250 and FID-50K@50 for comparison. As shown in Table 3, performance under the 50-step training schedule quickly saturates around 250 training steps and then steadily declines, whereas performance under the 250-step inference schedule continues to improve for another 200 training steps. This divergence highlights an adaptation bias toward the training denoising schedule, pointing to a underexplored characteristic of the denoising reduction paradigm that requires further investigation.

# B. Cross-Metric Evaluation

To verify that improvements from our distribution-wise reward training are not specific to the FID metric or the Inception-v3 feature space, we evaluate the fine-tuned SiT model (at 450 training steps) on a comprehensive set of independent metrics. As shown in Table 4, all metrics consistently improve, confirming genuine distributional improvement. Notably, $FD_{DINOv2}$ uses DINOv2 (Oquab et al., 2023) features entirely different from Inception-v3, providing strong evidence against Inception-specific exploitation. Precision and Density improvements indicate enhanced sample fidelity, while the modest Recall decrease and Coverage increase demonstrate that diversity is well preserved.

| Denoising Steps | | 50 | | 250 | |
|---|---|---|---|---|---|
| FID-#img | | 5K | 50K | 5K | 50K |
| | 0 | 20.92 | 13.78 | 14.54 | 8.86 |
| | 50 | 14.80 | 8.34 | 12.13 | 6.56 |
| | 100 | 13.55 | **7.57** | 13.09 | 7.12 |
| ← Training Steps | 250 | **13.30** | 7.73 | 12.57 | 6.81 |
| | 400 | 13.60 | 7.79 | 12.13 | 6.23 |
| | 450 | 14.15 | 7.93 | **11.48** | **5.77** |
| | 500 | 14.50 | 8.24 | 11.63 | 6.04 |

*Table 3.* The model exhibits an adaptation bias toward the training denoising schedule while training under denoising reduction paradigm. With 50 denoising steps for training and 250 for evaluation, performance with 50 steps saturated and worsened after 100 training steps, while 250-step performance remained improving.

*Table 4.* Cross-metric evaluation on SiT-XL/2 (ImageNet 256×256). All metrics are evaluated at 450 training steps using the same fine-tuned model. KID and MMD use Inception-v3 features with polynomial and Gaussian kernels respectively; $FD_{DINOv2}$ uses DINOv2 features.

| Metric | SiT Original | + Ours (RL) | Change |
|---|---|---|---|
| FID↓ | 8.30 | 5.77 | ↓30.5% |
| KID↓ | 0.0043 | 0.0020 | ↓53.5% |
| MMD↓ | 0.0029 | 0.0015 | ↓48.3% |
| $FD_{DINOv2}$ ↓ | 230.39 | 164.88 | ↓28.5% |
| Precision↑ | 0.6983 | 0.7286 | ↑4.3% |
| Recall↑ | 0.7527 | 0.7262 | −3.5% |
| Density↑ | 0.7673 | 0.8594 | ↑12.0% |
| Coverage↑ | 0.8698 | 0.8950 | ↑2.9% |

## C. Reward Variance Analysis

A potential concern with the subset-replace strategy is whether replacing only a small subset (e.g., 50 out of 5,000 images) introduces excessive noise in the reward signal. We provide a detailed variance analysis to demonstrate the stability of our reward computation.

**Overall reward stability.** Across 450 training steps, the reward coefficient of variation (CV) is 4.67%, and the intra-step FID CV caused by random replacement positions is only 0.14%. This indicates that the replacement position noise is negligible compared to actual sample quality differences.

**Variance across replacement sizes.** We measure the intra-step FID CV for different replacement subset sizes while keeping the reference set fixed at 5,000 images. As shown in Table 5, all CVs remain very low regardless of replacement size.

*Table 5.* Intra-step FID coefficient of variation (CV) for different replacement sizes. The low CV values confirm that the reward signal remains stable across all tested configurations.

| Replacement Size | 4 | 8 | 16 | 32 | 50 | 100 |
|---|---|---|---|---|---|---|
| FID CV (%) | 0.09 | 0.11 | 0.12 | 0.20 | 0.28 | 0.37 |

**Mechanisms bounding variance impact.** Three mechanisms prevent noisy reward estimates from destabilizing policy optimization: (1) best-of-N selection filters low-quality samples before replacement, (2) ratio clipping ($\varepsilon = 0.0001$) prevents large policy updates from any single step, and (3) advantage normalization standardizes the reward signal across the batch. Over the entire training, zero destructive policy updates were observed, and training exhibits stable, monotonic convergence.

## D. Computational Cost Analysis

We profile the per-step computational cost of our method on $8\times$ L40S GPUs to quantify the overhead introduced by the distribution-wise reward computation.

*Table 6.* Per-step computational cost breakdown for RL fine-tuning with the subset-replace strategy. The FID matrix computation is the only cost unique to our method; rollout generation and policy training are shared with any sample-wise RL approach.

| Component | Time (s) | Fraction (%) |
|---|---|---|
| Rollout generation | 22.7 | 10.3 |
| Policy training | 157.3 | 71.5 |
| FID matrix computation | 17.6 | 8.0 |
| Other (data loading, sync, etc.) | 22.4 | 10.2 |
| Pool regeneration (amortized) | – | 4.6 |

The FID matrix computation, which is the only component unique to our distribution-wise reward approach, accounts for only 8.0% of the total step time. The dominant costs—rollout generation (10.3%) and policy training (71.5%)—are shared with any RL fine-tuning method. Additionally, our reward model (Inception-v3, 24M parameters) is $12.7\times$ smaller than typical sample-wise reward models (e.g., CLIP ViT-L, 304M parameters), further reducing memory and compute overhead. The reference set is regenerated every 10 steps, adding approximately 4.6% amortized overhead.

## E. Limitations

Our current experiments focus on class-conditional ImageNet generation. Extending the subset-replace strategy to open-vocabulary text-to-image settings requires further exploration of how to construct representative reference sets without predefined class labels. Additionally, the reference set regeneration interval and size are currently tuned via ablation; an adaptive scheduling mechanism that adjusts these based on training dynamics could further improve efficiency and is left for future work.

## F. Qualitative Results

We visualize the image generation results of the pretrained SiT-XL/2 model and the model fine-tuned with distribution-wise reward RL from Section 4.1, as shown in Figures 6 to 10.

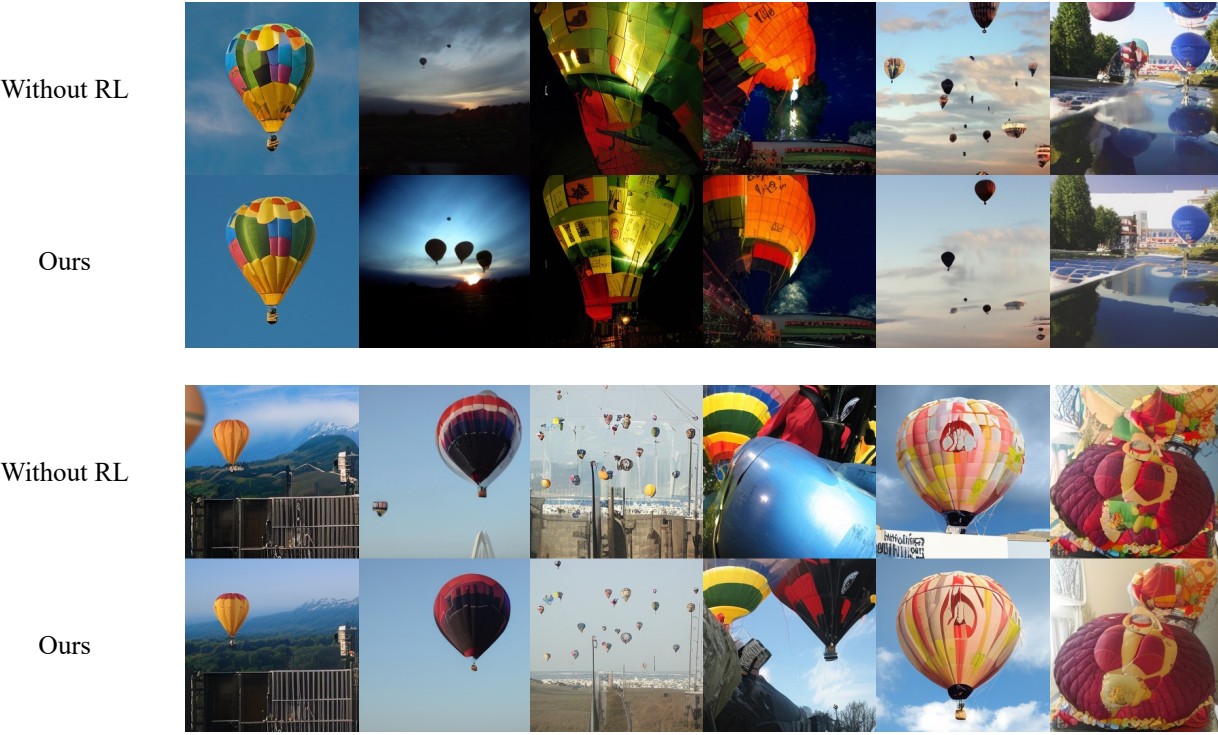

*Figure 6.* Uncurated samples of class label `"airliner"` (404)

*Figure 7.* Uncurated samples of class label `"balloon"` (417)

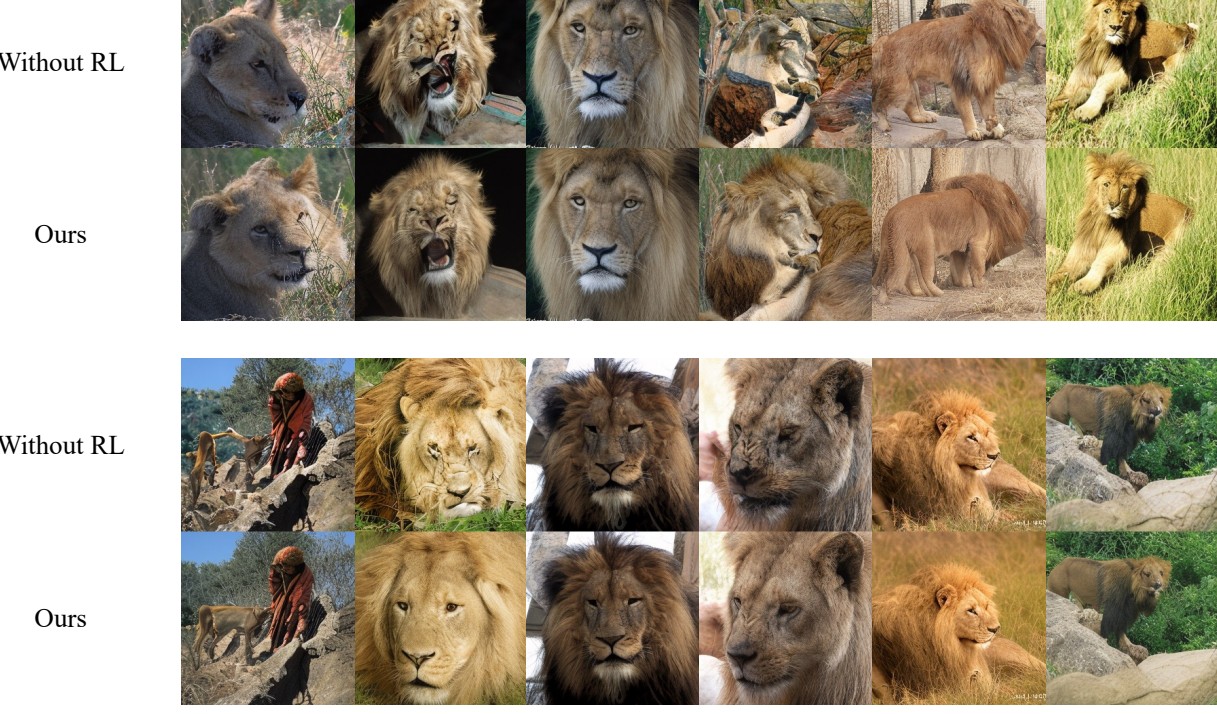

*Figure 8.* Uncurated samples of class label `"giant panda"` (388)

*Figure 9.* Uncurated samples of class label `"lion"` (291)

*Figure 10.* Uncurated samples of class label `"zebra"` (340)

