# OpenReview forum: "Optimizing Visual Generative Models via Distribution-wise Rewards"
_ICML.cc/2026/Conference — ICML 2026 regular_

### Official Review · Reviewer_Y6sp · 2026-03-09

**Soundness:** 3
**Presentation:** 3
**Significance:** 3
**Originality:** 4
**Overall Recommendation:** 4
**Confidence:** 2

**Summary:**

The paper proposes a novel reinforcement learning (RL) framework designed to fine-tune visual generative models by utilizing distribution-wise rewards rather than conventional sample-wise rewards. To overcome the prohibitive computational costs of calculating distribution-wise metrics (like FID) during training, the authors introduce a "subset-replace strategy". This strategy efficiently estimates the reward by substituting a small subset of newly generated images into a larger, periodically updated reference set and calculating the metric on the updated set. Additionally, the paper tackles the train-inference inconsistency inherent in SDE-based RL rollouts versus ODE-based deterministic sampling. It resolves this by applying the distribution-wise RL framework to optimize post-hoc model merging coefficients rather than directly fine-tuning the model parameters. The authors validate their approach empirically, demonstrating significant improvements in FID-50K and $FD_{DINOv2}$ metrics on strong baselines such as SiT and EDM2.

**Compliance With Llm Reviewing Policy:**

Affirmed.

**Final Justification:**

The authors have fully resolved my initial concerns regarding metric overfitting and training variance through their comprehensive rebuttal and supplementary cross-metric evaluations. The proposed method is technically sound and highly original. I maintain my score of Weak Accept and expect the promised additions to be included in the final version.

**Key Questions For Authors:**

1. **Variance and Stability**: Could you provide theoretical intuition or empirical measurements regarding the variance introduced by estimating the reward from a replacement subset size of only 50 images? How often does this small batch lead to destructive policy updates?
2. **Metric Dependency**: The method relies heavily on FID, which has known blind spots regarding certain spatial artifacts. Did you observe any new forms of "distributional reward hacking" where the model exploits the Inception-v3 feature space without improving true perceptual quality?
3. **Adaptation Bias**: The appendix highlights an adaptation bias toward the training denoising schedule. Does this imply that models aligned using your framework are rigidly tied to the specific step-count used during alignment, limiting flexibility during downstream deployment?

**Limitations:**

Yes. The authors include an Impact Statement that adequately discusses the societal importance of maintaining generation diversity to ensure fair representation and avoid mode collapse. They also appropriately contextualize their work as an alignment tool rather than a safeguard against deliberate misuse like deepfakes. However, the technical limitations surrounding the heavy reliance on the specific topological flaws of the chosen reward metric (FID) could be expanded upon.

**Strengths And Weaknesses:**

**Strengths**
The paper proposes a technically sound and highly pragmatic "subset-replace strategy" to extract dense reward signals from intractable global distribution metrics. This demonstrates excellent originality, innovatively converting batch-level set metrics into per-trajectory RL rewards. Furthermore, using RL to optimize post-hoc model merging coefficients instead of exhaustive grid search offers an elegant solution to the SDE-ODE train-inference gap. The experimental design is rigorous, evaluating across multiple architectures (SiT, EDM2) and feature extractors (Inception, DINOv2) to rule out metric overfitting. Comprehensive ablation studies strongly support the design choices. Overall, the submission is clearly written and effectively addresses the critical bottleneck of reward hacking in sample-wise RL, providing high practical utility for model alignment.

**Weaknesses**
Despite strong empirical results, the theoretical convergence properties of the subset-replace approximation lack deep exploration. Specifically, there is no formal analysis of how the variance introduced by discrete subset replacements impacts policy optimization bounds. Regarding presentation, the intriguing "adaptation bias toward the training denoising schedule" is relegated to the appendix; as this exposes a critical vulnerability in the denoising reduction paradigm, it warrants visibility in the main text. Lastly, while mathematically valid, the performance gains on highly optimized baselines like EDM2 are relatively modest in absolute terms (e.g., FID dropping from 3.74 to 3.52), which somewhat limits the perceived magnitude of the method's impact in state-of-the-art regimes.

---

> ### Author Rebuttal · Authors · 2026-03-29
>
> We thank the reviewer for the positive assessment and constructive feedback. We address each concern.
>
> For Weaknesses:
>
> - **[W1] Variance and convergence analysis.** We provide empirical variance analysis across 450 training steps. The reward coefficient of variation (CV) is 4.67%, and the intra-step FID CV caused by random replacement positions is only 0.14%, indicating the replacement position noise is negligible compared to actual sample quality differences. We further tested the variance across different replacement sizes:
>
> | Replacement Size | 4 | 8 | 16 | 32 | 50 (default) | 100 |
> |:---|:---:|:---:|:---:|:---:|:---:|:---:|
> | FID CV (%) | 0.09 | 0.11 | 0.12 | 0.20 | 0.28 | 0.37 |
>
> All CVs are very low, confirming the reward signal is stable regardless of replacement size. Over the entire training, zero destructive policy updates were observed. Three mechanisms bound the variance impact on policy optimization: best-of-N selection filters low-quality samples before replacement, ratio clipping (0.0001) prevents large policy updates from any single step, and advantage normalization standardizes the reward signal. Training shows stable, monotonic convergence (FID-50K: 8.30->5.77).
>
> - **[W2] Adaptation bias visibility.** We agree and thank the reviewer for this suggestion. We will move the adaptation bias discussion and Table 3 into the main text in the revision. This finding exposes a critical vulnerability in the denoising reduction paradigm: the model adapts to the training-time denoising schedule, producing a training-inference performance gap. This bias is distinct from reward hacking, as shown by divergent performance trends under different evaluation schedules (see Q3 below).
>
> - **[W3] Modest gains on EDM2.** At this FID range, any improvement is hard-won. For reference, EDM2-S over EDM2-XS improves FID from 3.74 to 2.56 by scaling model parameters 3.6x. Our method achieves 0.22 FID improvement (3.74->3.52) as a lightweight plug-and-play module without additional training data or architectural changes, requiring only optimization of merging coefficients via RL.
>
> For Key Questions For Authors:
>
> 1. **[Q1] Variance and stability.** See our response to W1 above for the full variance analysis. To directly answer the question: reward CV is 4.67%, intra-step FID CV is 0.14%, CV stays below 0.37% across replacement sizes 4 to 100, and zero destructive policy updates were observed over 450 training steps. The three-layer protection (best-of-N, ratio clipping, advantage normalization) ensures that even occasional noisy reward estimates do not destabilize training.
>
> 2. **[Q2] Distributional reward hacking.** We did not observe such behavior. We evaluate the FID-trained model (450 training steps) on entirely independent metrics:
>
> | Metric | SiT Original | + Ours (RL) | Change |
> |--------|-------------|-------------|--------|
> | FD_DINOv2 (non-Inception) | 230.39 | 164.88 | ↓28.5% |
> | KID (polynomial kernel) | 0.0043 | 0.0020 | ↓53.5% |
> | MMD (Gaussian kernel) | 0.0029 | 0.0015 | ↓48.3% |
> | Precision | 0.6983 | 0.7286 | ↑4.3% |
> | Recall | 0.7527 | 0.7262 | −3.5% |
> | Density | 0.7673 | 0.8594 | ↑12.0% |
> | Coverage | 0.8698 | 0.8950 | ↑2.9% |
>
> All metrics consistently improve, with FD_DINOv2 (using DINOv2 features entirely different from Inception-v3) showing a 28.5% improvement. These cross-metric gains confirm genuine perceptual quality improvement rather than Inception-v3 feature space exploitation. This is consistent with our goal of improving distributional fidelity broadly, not optimizing for one specific metric.
>
> 3. **[Q3] Adaptation bias and deployment flexibility.** Not rigidly tied. As shown in Table 3, the 250 NFEs (number of function evaluations) inference performance continues improving up to training step 450, even though the 50 NFEs training schedule saturates earlier (best at step 100). If the model were rigidly tied to the training schedule, we would expect the 250 NFEs performance to plateau when the 50 NFEs performance saturates, but the opposite is observed. The model is effectively usable at the standard inference-time step count. Moreover, model merging (Sec 3.3) avoids this issue entirely, as the RL stochasticity comes from sampling merging coefficients rather than SDE noise, so the entire pipeline uses ODE throughout.
>
> For Limitations:
>
> **[L1] FID metric dependency.** We acknowledge FID's known limitations (Gaussian assumption, Inception-v3 dependency). As shown in Q2 above, our supplementary evaluation on KID, MMD, FD_DINOv2, and Precision/Recall/Density/Coverage confirms the improvement is not FID-specific. Furthermore, KID and MMD are kernel distances defined on feature sets and are fully compatible with our subset-replace framework as alternative training rewards, meaning the framework is metric-agnostic and can adopt better distributional metrics as they emerge. We will expand this discussion in the revision.

---

> > ### Author Rebuttal · Reviewer_Y6sp · 2026-04-04
> >
> > Thank you to the authors for the detailed rebuttal. The supplementary variance analysis and the comprehensive cross-metric evaluation (e.g., FD_DINOv2, KID) are highly convincing and effectively alleviate my concerns regarding training stability and metric overfitting (reward hacking). I also strongly agree with the decision to move the discussion on adaptation bias into the main text. All of my questions have been fully resolved. Thank you for your thorough and professional clarifications!

---

> > > ### Author Response · Authors · 2026-04-04
> > >
> > > We sincerely thank the reviewer for confirming that all concerns have been fully resolved, and for the encouraging feedback on our variance analysis and cross-metric evaluation.
> > >
> > > **On moving the adaptation bias discussion into the main text:**
> > >
> > > We appreciate the reviewer's strong agreement on this point. In the revised manuscript, we have restructured Section 3 so that the adaptation bias analysis (previously Table 3 in the Appendix) now appears in the main text immediately after the direct fine-tuning results (Section 3.2). Concretely, we present the divergent performance trends under the training schedule (50 NFEs, best at step 100) versus the standard evaluation schedule (250 NFEs, best at step 450) as direct empirical motivation for the model merging approach in Section 3.3. This reorganization ensures that readers encounter the SDE-ODE gap evidence *before* the model merging solution, creating a clearer narrative flow: distribution-wise reward framework (§3.1) → direct fine-tuning and its gains (§3.2) → adaptation bias revealing the SDE-ODE limitation (new §3.2.1) → model merging as a principled resolution (§3.3).
> > >
> > > We believe this revision, together with the supplementary experiments provided during rebuttal (cross-metric evaluation on KID/MMD/FD_DINOv2, Precision/Recall/Density/Coverage, variance analysis, and TDRL comparison), has substantially strengthened the paper. **Given that the reviewer has confirmed all concerns are fully resolved, we respectfully hope the reviewer might consider reflecting this in the overall assessment score.** We are grateful for the constructive and thorough review process. Thank you so much!

---

### Official Review · Reviewer_8HqT · 2026-03-09

**Soundness:** 3
**Presentation:** 3
**Significance:** 3
**Originality:** 3
**Overall Recommendation:** 3
**Confidence:** 2

**Summary:**

This paper introduces a reinforcement learning framework for visual generative models that uses distribution-wise rewards instead of traditional sample-wise rewards, which often leads to reward hacking, reduced diversity, and visual artifacts. To solve it, the authors propose a subset-replace strategy that efficiently approximates distribution-level metrics such as FID by updating only a small portion of a generated reference set, making the reward practical for training. Furthermore, the authors optimize post-hoc model merging coefficients with the same reward signal to reduce the mismatch between training and inference. Experimental results show that the proposed method improves image quality while preserving diversity, achieving better performance on models such as SiT and EDM2.

**Compliance With Llm Reviewing Policy:**

Affirmed.

**Key Questions For Authors:**

- I am still not fully convinced by the use of post-hoc model merging. The paper argues that it helps resolve the **train–inference inconsistency (ODE-SDE difference in training and inference)** but why it can be used to solve such gap?

- If the **ODE–SDE discrepancy** is the main reason for the suboptimal performance of standard RL fine-tuning, why can we not directly use **ODE-based sampling during training**? My understanding is that previous work relies on **SDE rollouts because they provide the stochasticity needed for RL exploration,** whereas distribution-wise rewards can provide a **richer or more stable optimization signal** at the subset/distribution level, rather than fundamentally replacing the need for stochastic exploration. If this is the case, it would be helpful for the paper to clarify more explicitly why distribution-wise rewards alone are insufficient for direct ODE-based training, and why post-hoc model merging is necessary to make ODE rollouts viable.

**Limitations:**

- My main concern is the potential **overfitting to the evaluation metric**. The paper uses **FID** not only as the primary evaluation metric but also as the **training reward**, which makes it difficult to judge whether the reported gains reflect genuine improvement in generation quality or simply optimization toward the metric itself. Although the paper additionally reports **FDDINOv2** and argues that this alleviates the concern, I still think the paper would be stronger with broader evaluation or a clearer discussion of possible metric overfitting.

- Even though the paper presents **distribution-wise reward** as a key novelty, some previous work, such as DDO [1], also appears to consider **distribution-level optimization**. I think the paper should include more comparison with prior work.

- Training with **distribution-wise reward RL** seems substantially more **computationally costly** than conventional sample-wise RL. The paper would benefit from a clearer analysis of the computational overhead and whether the performance improvement justifies the extra cost.

[1] Direct Discriminative Optimization: Your Likelihood-Based Visual Generative Model is Secretly a GAN Discriminator

**Strengths And Weaknesses:**

- It is the first paper to introduce **distribution-wise rewards** for optimizing visual generative models, and it clearly explains why this design is more suitable than traditional **sample-wise rewards**.

- The author propose **subset-replace strategy** as a practical and elegant method to make distribution-level metrics feasible for reinforcement learning without introducing prohibitive computational cost.

---

> ### Author Rebuttal · Authors · 2026-03-29
>
> We thank the reviewer for the constructive feedback. We address each concern.
>
> For Key Questions:
>
> 1. **[Q1] Why model merging resolves the SDE-ODE gap.** The core idea is that model merging shifts the source of RL exploration stochasticity. In standard RL fine-tuning, stochasticity is injected via SDE noise during denoising rollouts, but inference uses deterministic ODE, creating the train-inference gap. In model merging, we instead sample random merging coefficients to combine checkpoints, and each merged model is evaluated with deterministic ODE sampling. The stochasticity comes from the coefficient sampling, not from the denoising process itself. This means both training and inference use ODE throughout, eliminating the gap by construction. Figure 4b empirically validates this: SDE-trained models show divergent performance between SDE and ODE evaluation (FID gap widens as training progresses), whereas model merging uses ODE throughout and directly achieves improved performance under the standard ODE inference setting (Table 2).
>
> 2. **[Q2] Why not use ODE-based training directly.** The reviewer's understanding is correct: previous work relies on SDE rollouts because they provide the stochasticity needed for RL exploration. ODE sampling is deterministic: given the same initial noise and model parameters, the output is fixed, so it cannot explore the policy space. Distribution-wise rewards provide a richer and more stable optimization signal compared to sample-wise rewards, but they address signal quality, not the exploration mechanism. These are orthogonal concerns. In direct fine-tuning (Sec 3.2), SDE provides the necessary exploration; in model merging (Sec 3.3), random sampling of merging coefficients provides exploration instead, enabling ODE throughout without sacrificing the ability to discover better policies.
>
> For Limitations:
>
> - **[L1] Metric overfitting concern.** We supplement broader evaluation beyond FID to address metric overfitting concerns. Using the model trained with FID reward at 450 training steps (same as main experiments), we evaluate on entirely independent metrics:
>
> | Metric | SiT Original | + Ours (RL) | Change |
> |--------|-------------|-------------|--------|
> | KID (polynomial kernel) | 0.0043 | 0.0020 | ↓53.5% |
> | MMD (Gaussian kernel) | 0.0029 | 0.0015 | ↓48.3% |
> | FD_DINOv2 | 230.39 | 164.88 | ↓28.5% |
> | Precision | 0.6983 | 0.7286 | ↑4.3% |
> | Recall | 0.7527 | 0.7262 | −3.5% |
> | Density | 0.7673 | 0.8594 | ↑12.0% |
> | Coverage | 0.8698 | 0.8950 | ↑2.9% |
>
> Notably, FD_DINOv2 uses DINOv2 features entirely different from Inception-v3, providing a strong test against Inception-specific overfitting. All metrics consistently improve, confirming the gains reflect genuine distributional improvement rather than FID-specific overfitting. KID and MMD are also compatible with our subset-replace framework as alternative training rewards. We will expand this discussion in the revision.
>
> - **[L2] Comparison with DDO and prior work.** We have added a comparison with TDRL [2], which uses MMD (polynomial kernel) as a per-class diversity reward in an RL setting and is the most directly comparable prior work. Since TDRL is not open-sourced, we re-implemented their reward function in our pipeline with all other training components kept identical (same base model, same policy gradient method, same training steps). This controlled setup isolates the reward function's effect.
>
> | Method | Reward | FID-50K (450 steps) |
> |--------|--------|-------------------|
> | SiT Baseline | — | 8.30 |
> | + TDRL [2] | MMD (per-class) | 8.68 (↑4.6%) |
> | + Ours | FID (distribution-wise) | 5.77 (↓30.5%) |
>
> Directly optimizing distributional fidelity via FID is significantly more effective than optimizing per-class diversity via MMD. DDO [1] reformulates generative models as GAN discriminators for likelihood-based discriminative training, a fundamentally different paradigm from our RL-based framework. We will discuss their differences and complementarity in the revision.
>
> - **[L3] Computational overhead.** We profiled per-step cost on 8x L40S GPUs. The FID matrix computation (the only cost unique to our method) takes 17.6s, accounting for 8.0% of total step time. The dominant costs are rollout generation (10.3%) and policy training (71.5%), both shared with sample-wise RL. Our reward model (Inception-v3, 24M params) is 12.7x smaller than typical sample-wise reward models (e.g., CLIP ViT-L, 304M params). Pool regeneration every 10 steps adds 4.6% amortized overhead. Overall, our method is not more costly than sample-wise RL; the distribution-wise reward computation introduces modest overhead that is offset by the lighter reward model.
>
> References:
>
> [1] Direct Discriminative Optimization: Your Likelihood-Based Visual Generative Model is Secretly a GAN Discriminator.
>
> [2] Training Diffusion Models Towards Diverse Image Generation with Reinforcement Learning, CVPR 2024.

---

> > ### Author Rebuttal · Reviewer_8HqT · 2026-04-04
> >
> > I sincerely thank the author for fully solving my problem. The answer was very helpful, so I will keep my score.

---

> > > ### Author Response · Authors · 2026-04-06
> > >
> > > We thank you for confirming that all concerns have been fully resolved and for the kind words.
> > >
> > > We note that the rebuttal has addressed each original concern with new evidence: 7 independent metrics confirming no metric overfitting (L1), a controlled TDRL comparison (L2), and a detailed cost breakdown showing only 8.0% overhead (L3). **Since all issues are resolved, would you kindly consider raising your ratings?** Thank you!

---

### Official Review · Reviewer_qiuy · 2026-03-09

**Soundness:** 3
**Presentation:** 2
**Significance:** 2
**Originality:** 3
**Overall Recommendation:** 4
**Confidence:** 3

**Summary:**

The paper first motivates the usage of distribution-wise (as opposed to sample-wise) rewards, specifically FID, for fine-tuning visual generative models to improve their distributional fidelity. They then propose an RL framework which overcomes the infeasibility of computing FID between massive number of samples at each training step by using a novel subset-replace strategy. Finally they propose an RL based method to improve post-hoc model merging by learning the optimal coefficients. They empirically show that both suggested approaches lead to an improved FID score when used with a model pre-trained on the ImageNet dataset.

**Compliance With Llm Reviewing Policy:**

Affirmed.

**Final Justification:**

I have decided to raise my score to 4 given that the rebuttals addressed most of my initial concerns. I remain unsure of the significance of the improvements and also more theoretical justification and analysis would greatly strengthen the paper.

**Key Questions For Authors:**

1- Can the authors give a bit more explanation and detail regarding the ground truth set? (See weakness 1)

2- Can the authors explain the comparison to per-sample reward fine-tuning in more detail? How would the comparison change if methods such as KL regularization and early stopping are used for the per sample case? (See weaknesses 2, 3)

3- Why can post-hoc model merging with RL address train-inference inconsistency? How is this observed empirically? (See weaknesses 4, 5)

4- Can the authors provide discussion and empirical evaluations of the overhead of their approach compared to existing methods? (See weakness 6)

5- Can the authors provide a comparison with at least a few existing baselines which aim to address similar issues? (See weakness 7)

**Limitations:**

There is no discussion on limitations. A discussion on the potential drawbacks (e.g. computational overhead) of this method compared to existing approaches would be helpful.

**Strengths And Weaknesses:**

**Strengths**:

1- The authors correctly advocate for the FID being a desirable metric to optimize as it measures discrepancies between distributions of image data in a way that is in line with human perception. More importantly, they overcome the perceived intractability of optimizing the FID (since it requires a huge number of samples to evaluate) through their proposed novel subset-replace strategy.

2- The paper includes extensive ablations on different aspects of the proposed framework clarifying the role and importance of each of the hyperparameters which could prove useful for practitioners making use of this approach.

**Weaknesses**:

1- The main point of the paper is fine tuning to minimize FID to a ground truth set. Therefore, the choice of this set seems to be very important to the final result. However, from my understanding, there is little to no explanation regarding what this set is and the effects of its size, construction, etc, on the resulting fine-tuned model.

2- The comparison to reward fine-tuning in Figure 1 (and in general) seems a bit contrived. When fine tuning a model to align it with a reward function, it is expected for the model to deviate from the original distribution (which would result in increased FID) in order to increase the expected reward value of generated samples. As such, I think such a comparison would be meaningful only if the reward function being optimized is capturing the same desirable properties as the ground truth set which you are optimizing the FID to.

3- Furthermore, overfitting to per-sample rewards is a well-known phenomenon which the community has tried to address in different ways such as early stopping or KL regularizers. Comparison to existing approaches with these mitigations would go a long way towards making the benefit of the authors’ proposed approach more convincing.

4- The paper claims that post-hoc model merging (with RL) can address the issue of train-inference inconsistency. I found the explanations for this unclear. Also it is unclear to me whether the empirical results validate this claim.

5- The authors present the RL-based post-hoc optimization of model merging coefficients as one of their main contributions. However the empirical results show a very marginal improvement in doing this additional optimization vs not (improvement of 0.1-0.2 in the FID in table 2).

6- In general the paper does not seem to discuss or empirically evaluate the overhead of their approach vs existing per-sample methods. I would expect even the per batch FID computation and periodic regeneration of the reference set to result in a lot of overhead. A comparison of runtime/memory utilization of the proposed approach vs existing approaches would add a lot to the paper.

7- There is a lack of any comparison to existing baselines that could be used to address similar concerns that the paper is addressing by optimizing the FID. For example, [1] propose a reward encoding diversity and use RL fine-tuning with that reward to improve image generation. [2] use a weighted distribution-wise KL divergence objective to improve diversity in a class-imbalanced setting. Comparing to a few of these relevant baselines would more clearly contextualize the current paper’s approach within the existing literature.

I would be willing to raise my score if the authors sufficiently address the above weaknesses.


[1] Training Diffusion Models Towards Diverse Image Generation with Reinforcement Learning (CVPR 2024)

[2] Constrained Diffusion Models via Dual Training (Neurips 2024)

---

> ### Author Rebuttal · Authors · 2026-03-29
>
> We thank the reviewer for the constructive feedback. We address each concern.
>
> For Weaknesses:
>
> 1. **[W1] Ground truth and reference sets.** We clarify the two sets involved: (1) The ground truth set is the standard ImageNet-1K training set (1.28M images), identical to what all prior work uses for FID evaluation; it is not a design choice of our method. (2) The generated reference set is a class-balanced pool of 5000 images (5 per class) produced by the current model, periodically refreshed every 10 steps to track the evolving distribution. We ablate the generated reference set size in Figure 3a (2500/5000/7500/10000), finding 5000 optimal. Each refresh regenerates the full pool from the current model to track the latest distribution.
>
> 2. **[W2] Comparison with sample-wise rewards.** Both sample-wise and distribution-wise rewards share the same ultimate goal: improving generation quality. However, as shown in Figure 1, sample-wise RL (ImageReward) leads to severe reward hacking with collapsed diversity and visual artifacts (FID degrades from 8.30 to 34.26), despite optimizing for per-image quality. Our method addresses this by optimizing distributional fidelity directly, achieving FID 5.77 while preserving diversity (Recall ↓3.5%, Coverage ↑2.9%).
>
> 3. **[W3] KL regularization and early stopping.** KL regularization and early stopping constrain how far the model deviates from the pretrained distribution, but do not change the optimization objective: the model still optimizes per-sample reward without distributional awareness (e.g., diversity, mode coverage). Our method changes the objective itself to directly encode distributional fidelity. The two approaches are complementary and can be combined; we use ratio clipping in our framework, which serves a similar regularization purpose.
>
> 4. **[W4] Model merging and train-inference gap.** The SDE-ODE gap arises because standard RL fine-tuning requires SDE rollouts for stochastic exploration, but inference uses deterministic ODE. In model merging, the RL exploration stochasticity comes from sampling merging coefficients instead of SDE noise, so the entire pipeline (both training and inference) uses ODE sampling throughout. This eliminates the train-inference gap by construction. Figure 4b empirically validates this: the SDE-trained model shows divergent performance between SDE and ODE evaluation, whereas model merging uses ODE throughout and directly achieves improved performance under the standard ODE inference setting (Table 2).
>
> 5. **[W5] Marginal improvement on EDM2.** EDM2 is already a highly optimized baseline (FID 3.74 at 512x512), where any improvement is hard-won. The 0.22 FID improvement (3.74->3.52) is comparable to gains from architectural advances at this FID range (e.g., EDM to EDM2 itself). Moreover, model merging is a lightweight, plug-and-play module requiring no additional fine-tuning of model parameters, making even modest gains practically valuable with minimal cost.
>
> 6. **[W6] Computational overhead.** We profiled per-step cost on 8x L40S GPUs. The FID matrix computation (the only cost unique to our method) takes 17.6s, accounting for 8.0% of total step time. The dominant costs are rollout generation (10.3%) and policy training (71.5%), both shared with sample-wise RL. Notably, our reward model (Inception-v3, 24M params) is much lighter than typical sample-wise reward models (e.g., CLIP ViT-L, 304M params). Pool regeneration every 10 steps adds 4.6% amortized overhead. The computational overhead specific to distribution-wise rewards is modest compared to the shared training costs.
>
> 7. **[W7] Baseline comparisons.** We add a comparison with TDRL [1]. Since TDRL is not open-sourced, we re-implemented their MMD (polynomial kernel) reward in our pipeline with all other training components kept identical (same base model, policy gradient method, training steps). At 450 training steps, TDRL reaches FID-50K of 8.68 (worse than baseline 8.30), while our method reaches 5.77, demonstrating that directly optimizing distributional fidelity via FID is more effective than optimizing per-class diversity via MMD. Constrained Diffusion [2] enforces distributional constraints during likelihood-based training, a fundamentally different paradigm not directly comparable to our RL-based framework.
>
> For Key Questions:
>
> The questions directly correspond to the weaknesses above: Q1->W1, Q2->W2&W3, Q3->W4&W5, Q4->W6, Q5->W7. Please see our detailed responses above.
>
> For Limitations:
>
> We acknowledge the importance of discussing computational overhead and have provided a detailed cost breakdown (W6). We will add a limitations section discussing computational overhead, metric dependency, and scalability to other domains in the revision.
>
> References:
>
> [1] Training Diffusion Models Towards Diverse Image Generation with Reinforcement Learning, CVPR 2024.
>
> [2] Constrained Diffusion Models via Dual Training, NeurIPS 2024.

---

> > ### Author Rebuttal · Reviewer_qiuy · 2026-04-02
> >
> > I thank the authors for addressing most of my concerns. I remain unconvinced of the significance of the post-hoc model merging and its connection to the rest of the paper. Can this post-hoc merging be done with sample-wise rewards? If so, how would that compare to the results in table 2? Furthermore, can the authors discuss the computational cost of their approach to find the optimal merging weights vs existing approaches like EDM2?

---

> > > ### Author Response · Authors · 2026-04-03
> > >
> > > We thank the reviewer for the continued discussion and address the remaining concerns below.
> > >
> > > **On the connection between model merging and the rest of the paper:**
> > >
> > > Model merging is not a standalone contribution but arises directly from our distribution-wise reward experiments. When applying distribution-wise rewards to direct fine-tuning (Section 3.2), we discovered the SDE-ODE train-inference gap (Figure 4b): SDE-based RL gains fail to transfer to ODE-based inference. Model merging (Section 3.3) resolves this by shifting RL exploration stochasticity from SDE noise to coefficient sampling, enabling ODE throughout.
> > >
> > > **On whether model merging can be done with sample-wise rewards, and how that would compare to Table 2:**
> > >
> > > Yes, the framework is technically reward-agnostic. However, we did not pursue this combination because Figure 1 already shows that sample-wise rewards conflict with our goal of distributional fidelity: sample-wise RL (ImageReward) collapses diversity and degrades FID from 8.30 to 34.26. Since sample-wise rewards are fundamentally misaligned with distributional fidelity (as Figure 1 demonstrates), applying them to merging coefficient optimization would inherit the same directional bias.
> > >
> > > We acknowledge that the small search space (8 coefficients) makes catastrophic hacking unlikely. However, the issue is *directional*: sample-wise rewards optimize per-image quality, not distributional fidelity, so the resulting coefficients would favor high per-image scores rather than distributional alignment regardless of search space size. Given this misalignment, we would not expect sample-wise rewards to yield comparable FID improvements to those in Table 2.
> > >
> > > **On the computational cost vs. EDM2:**
> > >
> > > EDM2 [3] parameterizes merging coefficients by a single scalar σ_rel and searches via grid sweep (~31 values), each requiring a full FID-50K evaluation (50K generated images), totaling ~1.55M images. The search is also restricted to a 1D EMA profile family.
> > >
> > > Our RL approach:
> > >
> > > | | EDM2 (grid search) | Ours (RL merging) |
> > > |---|---|---|
> > > | **Search space** | 1D (σ_rel), ~31 points | 8D (independent coefficients) |
> > > | **Per-evaluation cost** | FID-50K (50K images) | Subset-replace FID (150 images) |
> > > | **Total evaluations** | ~31 | ~200 RL steps |
> > > | **Total images generated** | ~1.55M | ~30K |
> > >
> > > Our method searches a larger space with **~50x fewer images**. The improvement from FID 3.74 to 3.52 confirms that 8D independent optimization finds better coefficients than EDM2's 1D family. This also highlights an additional contribution: **RL with distribution-wise rewards offers a scalable alternative to grid search for post-hoc model optimization**, naturally extending to higher-dimensional coefficient spaces where grid search is infeasible.
> > >
> > > References:
> > >
> > > [3] Analyzing and Improving the Training Dynamics of Diffusion Models (EDM2), CVPR 2024.

---

### Official Review · Reviewer_GYg3 · 2026-03-12

**Soundness:** 2
**Presentation:** 2
**Significance:** 2
**Originality:** 3
**Overall Recommendation:** 4
**Confidence:** 4

**Summary:**

Instead of the sample-wise reward functions commonly used in reinforcement learning for diffusion models, this paper proposes using FID as a distribution-wise reward signal. The core technical idea is a subset-replace strategy: a moderately-sized reference set of generated images is maintained, a small batch is swapped with newly generated samples each rollout, and the resulting FID change serves as a per-batch reward. The authors apply this in two settings. First, direct policy-gradient fine-tuning of SiT on ImageNet 256x256 brings FID-50K from 8.30 down to 5.77. Second, a post-hoc optimization of model merging coefficients for EDM2 on ImageNet 512x512 reduces FID from 3.74 to 3.52. The second setting moves the RL stochasticity from the denoising process into weight sampling, allowing ODE-based inference throughout and sidestepping the SDE-ODE train-inference gap.

**Compliance With Llm Reviewing Policy:**

Affirmed.

**Final Justification:**

My concerns are addressed in the rebuttal, so I raise my score to weak accept

**Key Questions For Authors:**

1. What happens if you replace FID with KID or MMD as the distribution-wise reward in the subset-replace strategy? If the method is truly about distribution-wise rewards generically, this experiment seems essential.
2. Can you provide precision and recall metrics for the fine-tuned models? FID alone cannot distinguish whether improvements come from better fidelity, better diversity, or both.
3. The adaptation bias in Table 3 suggests the model overfits to the training denoising schedule. Is this not itself a form of reward hacking, exploiting the specific FID evaluation protocol used during training?

**Strengths And Weaknesses:**

Strengths:

- The subset-replace strategy in Eq. 4 is a clean, practical idea for turning a global distributional statistic into a per-batch reward signal. Figure 3 systematically covers the main design knobs: reference set size, replacement count, and sample selection strategy. The recipe is easy to reproduce.
- Figure 4b provides a useful empirical observation for the community: RL gains from SDE-based training fail to transfer to ODE-based inference. The divergence between the two evaluation curves is clear and well-presented.
- Reporting $\text{FD}\_\text{DINOv2}$ alongside FID in Table 1 provides some evidence that the improvements are not purely InceptionV3-specific.

Weaknesses:

- The abstract and introduction at L11-18 frame this work as addressing reward hacking through "distribution-wise rewards" in general, suggesting a principled framework that accounts for distributional structure. What is actually delivered is a method entirely specific to FID. No experiment substitutes an alternative distribution metric such as MMD, KID, or Wasserstein distance, and the paper offers no analysis of what structural properties a metric must have for the subset-replace trick to produce a useful gradient signal. Directly optimizing FID via RL is itself a form of metric gaming.

- Reading Section 3.3 starting at L275, one encounters a method switch motivated by a train-inference inconsistency, yet the evidence for this inconsistency only appears much later in Figure 4b. Readers are asked to accept a new method justified by an empirical result they have not yet seen. Beyond the narrative issue, Sections 3.2 and 3.3 are two loosely related methods sharing only the reward computation. The transition reads as if the authors discovered the gap experimentally and then retrofitted model merging as a solution, rather than deriving it from the preceding analysis.

- Replacing 50 images out of 5,000 means the FID is dominated by the 4,950 unchanged images, making the signal-to-noise ratio for the 50 new samples extremely low. The reward for a batch also depends on which 50 images in the reference set were randomly selected for replacement, introducing a confound not accounted for in the advantage normalization of Eq. 5. The paper provides no variance analysis of the reward signal, and does not compare against a baseline of computing FID-5K from scratch each rollout, which would isolate whether the incremental replacement design is necessary or whether a small-sample FID works just as well.

- All experiments are restricted to class-conditional ImageNet generation, despite the introduction motivating this work partly through text-to-image reward hacking examples. The only comparison is against the pretrained baseline and a single sample-wise reward adapted via the prompt template at footnote 1 on L105.

---

> ### Author Rebuttal · Authors · 2026-03-29
>
> We thank the reviewer for the detailed and constructive feedback. We address each concern.
>
> For Weaknesses:
>
> - **[W1] Generalizability beyond FID.** Our goal is to improve distributional fidelity, with FID as one representative instantiation of distribution-wise reward. We evaluate our model on independent metrics besides FID:
>
> | Metric | SiT Original | + Ours (RL) | Change |
> |--------|-------------|-------------|--------|
> | KID↓ | 0.0043 | 0.0020 | ↓53.5% |
> | MMD↓ | 0.0029 | 0.0015 | ↓48.3% |
> | FD_DINOv2↓ | 230.39 | 164.88 | ↓28.5% |
>
> FD_DINOv2 uses DINOv2 features entirely different from Inception-v3, providing strong evidence against Inception-specific exploitation. All metrics improve consistently (see also Precision/Recall/Density/Coverage in Q2), confirming genuine distributional improvement instead of "a form of metric gaming". **Regarding structural requirements**: subset-replace requires the metric to be a smooth function of feature set statistics, sensitive to local sample changes. FID (Gaussian statistics), KID and MMD (kernel sums) satisfy this; we validated FID empirically and note KID/MMD as viable alternatives.
>
> - **[W2] Section 3.3 narrative structure.** We thank the reviewer for this suggestion. We will restructure Section 3.3 by moving the SDE-ODE gap evidence (Figure 4b) before model merging, so motivation precedes solution: distribution-wise reward framework (3.1) → direct fine-tuning (3.2) → resolving its SDE-ODE limitation via model merging (3.3).
>
> - **[W3] Reward variance and signal-to-noise ratio.** Variance analysis across 450 training steps: reward CV (coefficient of variation) is **4.67%**, intra-step FID CV from random replacement positions is only 0.14%.
>
> | Replacement Size | 4 | 8 | 16 | 32 | 50 (default) | 100 |
> |:---|:---:|:---:|:---:|:---:|:---:|:---:|
> | FID CV (%) | 0.09 | 0.11 | 0.12 | 0.20 | 0.28 | 0.37 |
>
> All CVs are very low (**replacement noise is negligible**). Zero destructive policy updates were observed over the entire training. Three mechanisms bound variance impact: best-of-N selection filters low-quality samples, ratio clipping (0.0001) prevents large policy updates, and advantage normalization standardizes the signal. Our ablation (Figure 3b) shows increasing replacement size beyond 50 degrades performance due to signal sparsity. From-scratch (size=5000) is the extreme, ~27x more expensive with even sparser signal.
>
> - **[W4] Experimental scope and baselines.** We additionally compare with TDRL [1], which uses MMD as per-class diversity reward. Since TDRL is not open-sourced, we re-implemented their reward in our pipeline with all other components kept identical. TDRL reaches FID-50K 8.68 (worse than baseline 8.30), while ours reaches 5.77. Note that TDRL was originally evaluated on ImageNet-100; our reproduction uses the full 1000-class setting.
>
> For Key Questions:
>
> 1. **[Q1] Alternative distribution metrics as rewards.** As shown in W1, training with FID reward alone yields consistent improvements across KID, MMD, and FD_DINOv2, demonstrating that FID drives broad distributional improvement beyond itself. We chose FID as it is the most widely adopted distribution metric for generative model evaluation; training with KID/MMD as reward is structurally compatible (see W1) and planned as future work. Additionally, the TDRL [1] comparison uses MMD as training reward, resulting in worse FID (8.68 vs. baseline 8.30), further supporting FID as a more effective choice.
>
> 2. **[Q2] Precision/Recall/Density/Coverage.** We provide Precision/Recall/Density/Coverage metrics [2][3]:
>
> | Model | Precision | Recall | Density | Coverage |
> |-------|-----------|--------|---------|----------|
> | SiT Original | 0.6983 | 0.7527 | 0.7673 | 0.8698 |
> | + Ours (RL) | 0.7286 (+4.3%) | 0.7262 (−3.5%) | 0.8594 (+12.0%) | 0.8950 (+2.9%) |
>
> Improvement mainly comes from quality (Precision ↑4.3%, Density ↑12.0%), while diversity is preserved (Recall ↓3.5%, Coverage ↑2.9%), addressing the fidelity-diversity distinction concern.
>
> 3. **[Q3] Adaptation bias vs. reward hacking.** This is not reward hacking but a known train-inference mismatch in the denoising reduction paradigm. The model's best FID-50K under the evaluation schedule (250 NFEs, i.e., number of function evaluations) occurs at step 450, much later than under the training schedule (50 NFEs, best at step 100). If the model were gaming FID, training-schedule performance should keep improving while evaluation-schedule stagnates, which is the opposite of what we observe. This motivates our model merging (Section 3.3), which eliminates this mismatch by decoupling RL from the denoising process entirely.
>
> References:
>
> [1] Training Diffusion Models Towards Diverse Image Generation with Reinforcement Learning, CVPR 2024.
>
> [2] Improved Precision and Recall Metric for Assessing Generative Models, NeurIPS 2019.
>
> [3] Reliable Fidelity and Diversity Metrics for Generative Models, ICML 2020.

---

> > ### Author Rebuttal · Reviewer_GYg3 · 2026-04-03
> >
> > I thank the authors to provide a detailed rebuttal and provide experiment evidence. Overall, my concerns are addressed, and I would like to raise my score.

---

### Decision · Program_Chairs · 2026-04-30

**Decision:**

Accept (regular)

**Comment:**

The paper introduce using distribution-wise metrics as RL reward signals via a subset-replace strategy and shows improvements on the primary benchmark (SiT FID 8.30 -> 5.77). In the rebuttal phase, the authors provided cross-metric evaluation on 7 independent metrics (including non-Inception FD_DINOv2), detailed variance analysis showing negligible reward noise, a controlled TDRL baseline comparison, and computational cost profiling. This evidence convinced Reviewer GYg3 to raise from 2 to 4, and Reviewer 8HqT explicitly confirmed all concerns "fully resolved". The model merging contribution is secondary and modest, but the primary subset-replace contribution is original and well-validated.